# Age at Glaucoma Diagnosis in Germline Myocilin Mutation Patients: Associations with Polymorphisms in Protein Stabilities

**DOI:** 10.3390/genes12111802

**Published:** 2021-11-16

**Authors:** Tarin Tanji, Emily Cohen, Darrick Shen, Chi Zhang, Fei Yu, Anne L. Coleman, Jie J. Zheng

**Affiliations:** Stein Eye Institute, Department of Ophthalmology, David Geffen School of Medicine at UCLA, Los Angeles, CA 90095, USA; tarintaeko@gmail.com (T.T.); emilycohen96@gmail.com (E.C.); shendarrick821@gmail.com (D.S.); czhang@jsei.ucla.edu (C.Z.); fyu@ucla.edu (F.Y.); coleman@jsei.ucla.edu (A.L.C.)

**Keywords:** trabecular meshwork, myocilin, intraocular pressure, endoplasmic reticulum stress

## Abstract

Glaucoma is the leading cause of irreversible blindness worldwide, with elevated intraocular pressure (IOP) as the only known modifiable risk factor. Trabecular meshwork (TM)-inducible myocilin (the *MYOC* gene) was the first to be identified and linked to juvenile and primary open-angle glaucoma. It has been suggested that mutations in the *MYOC* gene and the aggregation of mutant myocilin in the endoplasmic reticulum (ER) of TM may cause ER stress, resulting in a reduced outflow of aqueous humor and an increase in IOP. We selected 20 *MYOC* mutations with experimentally determined melting temperatures of mutated myocilin proteins. We included 40 published studies with at least one glaucoma patient with one of these 20 *MYOC* mutations and information on age at glaucoma diagnosis. Based on data from 458 patients, we found that a statistically significant but weak correlation was present between age and melting temperature based on various assumptions for age. We therefore conclude that genetic analysis of *MYOC* mutations alone cannot be used to accurately predict age at glaucoma diagnosis. However, it might be an important prognostic factor combined with other clinical factors for critical and early detection of glaucoma.

## 1. Introduction

Glaucoma is the leading cause of irreversible blindness and is characterized by optic nerve damage and retinal ganglion cell loss [1,2]. Glaucoma affects approximately 70 million people worldwide, and is projected to affect 111.8 million people by 2040 [3]. Risk factors include a family history of glaucoma, age, chronic steroid use, myopia, diabetes, and hypertension [4]. Primary open-angle glaucoma (POAG) accounts for 90 percent of glaucoma cases and is defined by an open anterior chamber angle and elevated intraocular pressure (IOP) [3,5,6]. IOP is a measurement of the aqueous humor produced by the ciliary bodies that is drained out through the trabecular meshwork (TM) and uveoscleral outflow [6]. The TM, located in the iridocorneal angle, is a specialized and dynamic tissue that regulates IOP [6,7]. Changes in the cellular signaling pathways and physical structure of TM may increase resistance to aqueous humor outflow, which can increase IOP [8]. Increased IOP may then damage the optic nerve and cause irreversible glaucomatous optic nerve damage and subsequent vision loss [2]. Visual outcomes of patients with glaucoma are time-dependent, so improving current diagnostic methods for earlier detection and treatment of glaucoma is critical for minimizing the progression of irreversible vision loss [1,4].

Genetic analysis of glaucomatous mutations may be helpful in the early detection of glaucoma. Myocilin (the *MYOC* gene) was the first protein to be identified and linked to both juvenile open-angle glaucoma (JOAG) and POAG [9]. Myocilin is expressed in the TM tissues, and myocilin expression is more extensive in the TM of patients with POAG compared to those who do not have glaucomatous phenotypes [10,11,12,13]. However, the function of myocilin in POAG development remains unknown. Nevertheless, approximately 3% to 5% of the 70 million people affected by glaucoma have *MYOC* mutations [10,14]. The major component of the myocilin protein is a well-structured olfactomedin (OLF) domain, and more than 90% of glaucoma-related mutations are located in this domain [12,15]. Therefore, it has been suggested that *MYOC* mutations and the aggregation of mutant myocilin in the ER of TM may cause cell stress, resulting in reduced cell function, decreased outflow of aqueous humor, and increased IOP, a glaucoma risk factor [16,17,18,19]. Indeed, a transgenic mouse model with the *MYOC*Y437H mutation, demonstrating chronic ER stress, was associated with elevated IOP and TM cell death; however, administration of the chemical chaperone phenylbutyric acid decreased both myocilin accumulation in the ER and ER stress [20]. The conjugate base of the same chemical chaperone, sodium phenylbutyrate, also relieved ER stress in a transgenic mouse model [21]. ER stress was also relieved by knocking down expression of mutant *MYOC* using CRISPR-Cas-9-based treatment [22].

The stability of a protein can be defined by the protein’s melting temperature—the temperature at which fifty percent of the tested protein is denatured. The melting temperatures of the wild-type and several glaucoma-associated mutated OLF domains of myocilin were experimentally determined [23,24]. Indeed, compared to the wild type, the myocilin mutants responsible for glaucoma do have lower melting temperatures. Moreover, a moderate correlation (correlation coefficient of 0.54) between the ages at glaucoma diagnosis of patients with *MYOC* mutations with the melting temperatures of the mutated myocilin OLF domains was identified [23], further suggesting that the ER stress associated with aggregation of mutant myocilin plays a key role in IOP elevation. However, this notion has been challenged. For example, the effects of the most common *MYOC* variant (p.Gln368Ter) on IOP in several large-scale population panels were studied, and the results of those studies are not consistent [25,26,27]. This mutant loses about half of the myocilin OLF domain, and the rest of the OLF is completely unfolded [23]. However, it was found that the penetrance of this mutation is relatively lower in different populations [26,27], although a later study suggests that the lower penetrance may be caused by underdiagnosis [28].

To further evaluate this issue, we decided to examine all the literature that reported on glaucoma patients with *MYOC* mutations and to assess the relationship between the melting temperature of mutated myocilin proteins and the age at glaucoma diagnosis of patients through a meta-analysis. A correlation between age at diagnosis of glaucoma and *MYOC* mutations can inform patients with these *MYOC* mutations whether they are at a higher risk of developing glaucoma before irreversible vision loss occurs. We confirmed the correlation between the melting temperatures of glaucoma-associated myocilin mutants and age at glaucoma diagnosis [23,24], but the correlation we obtained is weak. While this weak correlation would validate the notion that myocilin protein stability is associated with TM cell death due to ER stress, it may also suggest that other factors, such as environment and genetic background, may also play a role in glaucoma development [29]. Indeed, it is known that the prevalence of glaucoma increases with age [3,30].

## 2. Materials and Methods

### 2.1. Eligibility Criteria for Considering Studies

The study included melting temperatures of the wild-type myocilin protein and of 22 mutated myocilin proteins that have been experimentally measured. Nineteen of the mutants had lower melting temperatures than that of the wild-type and three mutants had higher melting temperatures [24]. Among those 22 *MYOC* mutations, we found 20 mutations on the myocilin.com website [15,31] associated with at least one glaucoma patient (the last search was on 1 May 2018). The 20 mutations include the 19 mutations that have lower melting temperatures than the wild-type and the K398R mutant that has a slightly higher melting temperature than the wild-type (Table 1 and Figure 1).

Guided by the data present in myocilin.com [31] the literature was searched for publications from 1997 to 2018 that are associated with those 20 *MYOC* mutations. A total of 80 publications were identified. Each publication was reviewed for at least one patient with a *MYOC* mutation and age at diagnosis of glaucoma, and a total of 40 articles (listed in the Supplemental Information) met the following inclusion criteria: contains at least one patient with a *MYOC* mutation with a glaucoma diagnosis. The primary study outcome was the age at glaucoma diagnosis of patients with one of these 20 *MYOC* mutations.

### 2.2. Data Synthesis and Analysis

Publications in myocilin.com [31] were identified from December 2017 to May 2018. Several publications only describe aggregate mean or median ages of diagnosis for a group of patients who carry the same *MYOC* mutation. The authors of those publications were contacted. Because of different reasons, we were still unable to obtain individual ages of diagnosis of some patients. Nevertheless, for all the patients whose ages at glaucoma diagnosis could not be extracted, we obtained a mean or median of onset of those patients and the number of individuals with the *MYOC* mutation. Therefore, our data was compiled and sorted into two categories: data of aggregate mean or median ages of diagnosis for a group of patients with a single mutation or individual ages of diagnosis for patients with a given mutation. To reconcile the two types of data, the correlation between age at diagnosis and melting temperature was examined by two different methods.

The first method was a hot-deck analysis of the summary data, where n number of individuals for each mean age reported in each publication were included in a regression analysis. The number n refers to the number of individuals included in the reported mean. The second method imputed n individual ages drawn randomly from a Gaussian distribution, assuming mean and standard deviation as reported in each publication. For publications that did not provide standard deviation, an approximation of standard deviation was calculated as the range (maximum age–minimum age) divided by 4. In addition, since some random age values might be out of the reported range due to randomly being drawn from a Gaussian distribution, the random age values were truncated to the reported range when available.

Finally, the imputed age from the two imputation methods were combined with the reported individual age values for all publications, and the Pearson correlation coefficient between age and melting temperature was calculated for both sets. The scatter plots with a fitted linear line were generated to illustrate the correlation.

## 3. Results

The experimentally determined melting temperatures of 20 myocilin mutations were obtained from Donegan et al. [24]. A total of 80 published studies were identified; 40 publications met the inclusion criterion: at least one patient carrying one of the 20 *MYOC* mutations, and information on glaucomatous phenotypes. From those publications, we extracted a total of 458 patients (Table 1); among them, the number of patients whose age at glaucoma diagnosis were available is 283. There were also 175 patients whose individual age at glaucoma diagnosis was unknown; only the summary statistics for age at diagnosis of these patients were known. These patients carried the following nine *MYOC* mutations (I477N, I477S, Y427H, C433R, R272G, V426F, E323K, T377M, G364V). In an attempt to include these patients in our statistical analysis, we used two different methods. In the first method, we used hot-deck imputation [71] to include 175 patients (Figure 2). In the second method, we imputed the individual ages of these patients as a random value drawn from a Gaussian distribution, assuming the mean and standard deviation as reported in each publication (Figure 3).

The analysis results of the two methods including: the Pearson correlation coefficient, linear regression model, and scatter plot between age and melting temperature, are shown in Figure 2 and Figure 3. Similar results obtained from both methods of analysis gave us similar results. The correlation between age and melting temperature based on various assumptions for age (known individual age only or known individual age plus imputed age for those with unknown age) is statistically significant. The correlation coefficients calculated from both methods were 0.38 and 0.34, respectively, as evidenced by the wide range of age distribution at each melting temperature shown in the scatter plots, especially for the middle range of melting temperature (38 degrees to 45 degrees).

## 4. Discussion

ER stress implicated in ocular diseases including glaucoma and neurodegenerative diseases [17,72] can be induced by genetic mutations, overexpression of genes, or other pathophysiological processes that lead to protein aggregation in the ER lumen [73]. When large quantities of the misfolded myocilin aggregate in the ER lumen, the unfolded protein response (UPR) pathway can induce cell death [74,75]. Our analysis shows there is a weak to moderate correlation between the stability of mutated myocilin and age at glaucoma diagnosis of those patients who have the mutations. This is consistent with the hypothesis of the ER stress mechanism in glaucomatous TM [18,19,76].

Myocilin is overexpressed in TM cells, and its expression can be further enhanced by factors such as aging, mechanical stress, and steroid treatment. In a clinical setting, glucocorticoids are commonly used to treat postoperative inflammation, but may lead to steroid-induced ocular hypertension (SIOH) and steroid-induced glaucoma (SIG) [77]. However, it is unclear whether myocilin increases the risk of SIOH or SIG [77]. Treatment of one type of glucocorticoid, dexamethasone, was found to induce a significant increase in myocilin in cultured human TM cells, explants, and perfusion-cultured cells [78,79]; the range of induction (fold increase) of myocilin was 0.9–20 [78]. On the other hand, dexamethasone treatment resulted in a range of IOP increases and significant increases of at least 21 mmHg among only 30 percent of the participants [80]. It is likely that the dexamethasone-related IOP increase is connected to the dexamethasone-induced increase in myocilin protein expression [81]. Therefore, for those patients who carry *MYOC* missense mutations, myocilin overexpression may lead to additional ER stress and cell death [2,18,77].

However, ER-stress-induced URP can not only trigger cell death, but can also stimulate ER-associated protein degradation (ERAD) to prevent cell death [75,82]. The two opposite effects induced by ER stress, combined with the natural variation of myocilin production in the general population, may explain why there is a weak correlation between protein stability of mutated myocilin and age of glaucoma diagnosis. In our study, while we confirmed the positive correlation, the correlation we obtained is much weaker than what was reported previously, which is based on a limited data set [23]. Our study indicates that although many individuals with *MYOC* mutations have ER stress in the TM, not every individual with an *MYOC* mutation will respond with a stress-induced, high-fold increase in myocilin expression and develop myocilin-associated glaucoma [83]. This may be due to possible moderating variables, including varying patient environments and different genetic components. Indeed, it has been shown that diet, exercise routine, and patient lifestyle may affect IOP and influence POAG development [84,85,86]. For example, smoking and consuming coffee may increase IOP, while engaging in general physical exercise may decrease IOP [86,87]. Data from the National Health and Nutrition Examination Survey found that adult participants aged 40 years and older who performed moderate amounts of vigorous activity had 95% decreased odds of developing glaucoma compared to those who did not perform vigorous activity [86]. In the context of URP-mediated TM disruption, exercise reducing the prevalence of glaucoma could make sense, as exercise is shown to reduce oxidative stress and reverse age-related ER and mitochondrial dysfunction [88].

Despite weak variables, this correlation may be an important prognostic factor for the early detection of glaucoma and may be useful to future understanding of genetic tools in clinical care. The slopes calculated from the two methods are very similar in the linear relationship between age of diagnosis and the mutants’ melting temperature (1.58 obtained from the first method and 1.65 from the second method). If we take the average of the two values, 1.62, as the result of our study, this means that every degree decrease in melting temperature due to mutation lowers the age of diagnosis by 1.62 years. Glaucoma often begins asymptomatically [84,89], so it is critical to diagnose patients early before irreversible vision loss has already occurred. Equipped with such knowledge, physicians can more accurately determine which patients require increased screening for early detection of glaucoma, which is particularly critical for pediatric patients. Physicians could screen for patients with a family history of glaucoma and refer them for genetic testing at a clinical laboratory that meets the Clinical Labs Information Act standards. The presence of *MYOC* mutations would be clinically instructive for patients’ glaucoma treatment plans. Patients with *MYOC* mutations would undergo increased surveillance to detect any signs of glaucoma as early as possible, rather than the usual standard of eye care. The results of our study provide knowledge for patients with *MYOC* mutations concerning their risk for developing glaucoma in the future and may minimize irreversible vision loss through increased screening for patients at risk for glaucoma diagnosis.

Finally, other than the 20 *MYOC* mutations we examined in this study, there are many more known *MYOC* mutations that have been implicated in glaucoma [12,13]. It is likely that most of these mutations, if not all, are less stable than the wild-type myocilin, and that they cause ER stress in TM cells similar to the 20 mutations examined in this study [17,18]. Therefore, the correlations between the stabilities of these mutated myocilin proteins and age at glaucoma diagnosis of patients who carry the mutations should be similar as well. Further biophysical studies of the thermostability of those myocilin mutants, together with accurate genomic testing, will provide us with valuable information for glaucoma prevention and treatment.

## Figures and Tables

**Figure 1 genes-12-01802-f001:**
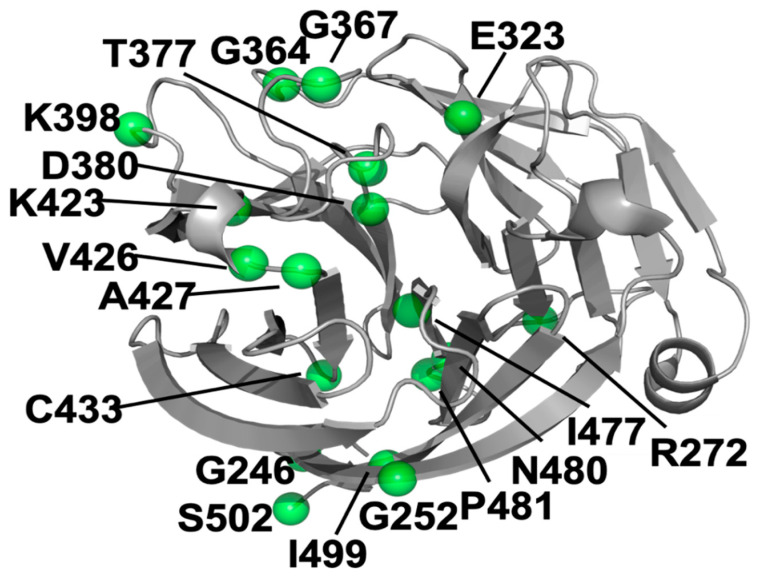
Cartoon structure of the OLF domain of myocilin. The amino acids with known mutations used for statistical analysis are shown as green balls.

**Figure 2 genes-12-01802-f002:**
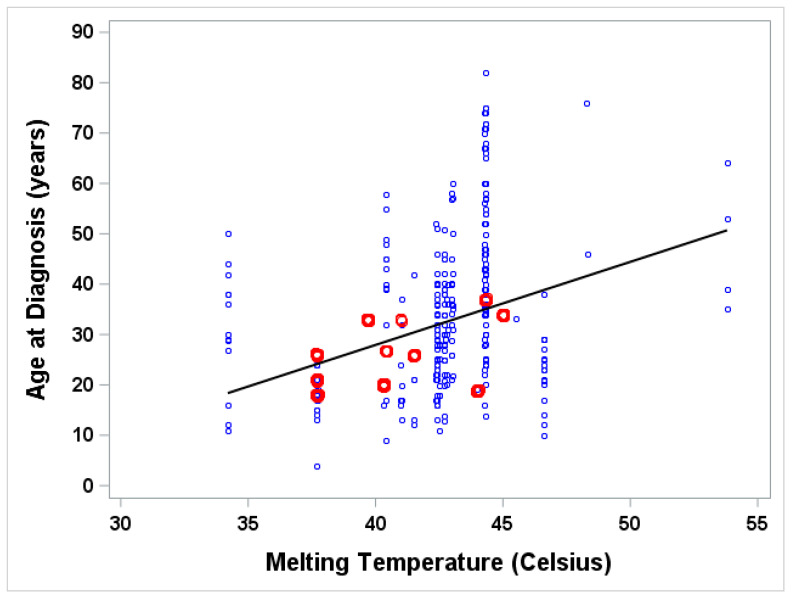
Plot of age at diagnosis versus the respective myocilin melting temperatures. Known individuals are blue; unknown individual ages at diagnosis are red. Pearson correlation coefficient between age of diagnosis and melting temperature is 0.37626. Linear regression of age of diagnosis (Age) on melting temperature (MT) is: Age = −38.05 + 1.65 × MT (*p* < 0.0001); R-squared = 0.142; *n* = 458.

**Figure 3 genes-12-01802-f003:**
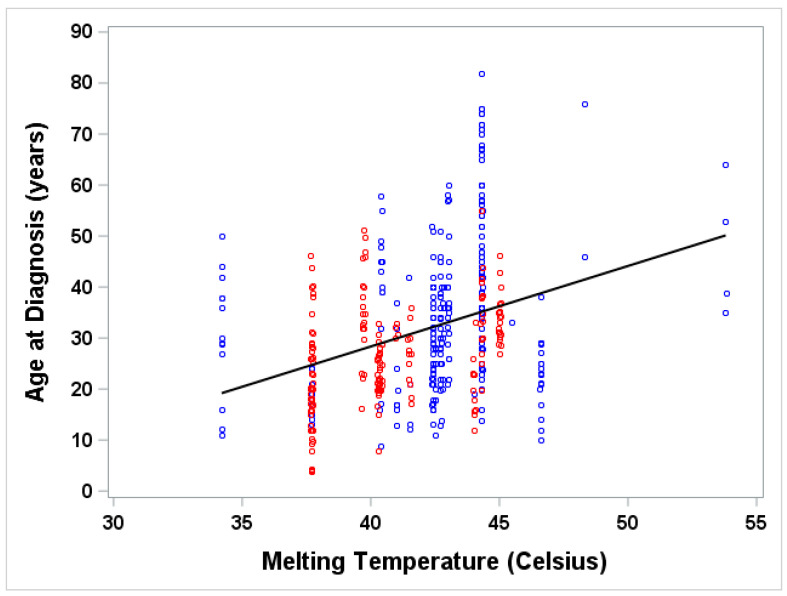
Age at diagnosis vs melting temperature, using randomly generated data points according to literature with summary statistics. Age for those with known individual ages (blue) and those with unknown individual ages (red) whose ages were imputed by a value randomly drawn from a Gaussian distribution with known mean age and standard deviation, SD, (SD was calculated as [maximum age–minimum age]/4 if SD was not given) and truncated by the given range. Pearson correlation coefficient between age of diagnosis and melting temperature is 0.34489. Linear regression of age of diagnosis (Age) on melting temperature (MT) is Age = −34.82 + 1.58 × MT (*p* < 0.0001); R-squared = 0.119; *n* = 458.

**Table 1 genes-12-01802-t001:** Number of Patients (known and unknown individual age).

MYOC	MT *	N (Known + Unknown) **	Frequency (%)
K423E [29,32,33]	34.2	13 (13 + 0)	2.84
I477N [34,35,36,37,38]	37.7	66 (16 + 50)	14.41
I477S [39]	39.7	20 (0 + 20)	4.37
Y427H [40]	40.3	35 (1 + 34)	7.64
C433R [41,42]	40.4	20 (13 + 7)	4.37
R272G [34]	41	5 (1 + 4)	1.09
S502P [43]	41	8 (8 + 0)	1.75
V426F [34,36,44]	41.5	17 (5 + 12)	3.71
N480K [39,45,46,47,48,49]	42.4	47 (47 + 0)	10.26
G246R [39]	42.5	7 (7 + 0)	1.53
G367R [32,37,50,51,52,53,54,55,56]	42.7	30 (30 + 0)	6.55
I499F [39]	42.8	7 (7 + 0)	1.53
G252R [34,36,37,50,57,58]	43	23 (23 + 0)	5.02
E323K [34,36]	44	12 (1 + 11)	2.62
T377M [34,35,37,40,51,59,60,61,62,63,64,65]	44.3	100 (85 + 15)	21.83
G364V [35]	45	22 (0 + 22)	4.80
P481L [32]	45.5	1 (1 + 0)	0.22
D380A [43,66]	46.6	19 (19 + 0)	4.15
A427T [67,68]	48.3	2 (2 + 0)	0.44
K398R [69,70]	53.8	4 (4 + 0)	0.87

* MT: Melting Temperature. ** N (Known + Unknown): Number of patients (number of patients with known individual on set age + number of patients with unknown individual on set age). Total numbers 458 (283 + 175).

## Data Availability

All the data obtained in the study are presented.

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
