# Peer review of "Age at Glaucoma Diagnosis in Germline Myocilin Mutation Patients: Associations with Polymorphisms in Protein Stabilities"

_genes, 2021, doi:10.3390/genes12111802_

Round 1

Reviewer 1 Report

The introduction could be rewritten to better explain why the study was initiated.  What is to be gained by having a correlation between age and MYOC mutations?

The materials and methods can start with the second paragraph. The first paragraph of this section still appears to be background. 

Author Response

We have modified the Introduction section (highlighted in yellow) as suggested by the reviewer. However, with due respect, we have decided not to move the first paragraph of the Methods section because we feel the arrangement makes the flow of the manuscript better.

Reviewer 2 Report

The paper entitled “Age at Glaucoma Diagnosis in Germline Myocilin Mutation Patients: Associations with Polymorphisms in Protein Stabilities” is a systematic review based on the importance of MYOC mutations in relation to melting temperatures of mutated myocilin proteins and age of diagnosis of glaucoma. The manuscript is well written and of clinical interest. The multifactorial aspect and limited current knowledge regarding the causes of glaucoma renders systematic reviews such as the one presented here of great interest. The role of genetic mutations of proteins found in the trabecular meshwork are useful in understanding physiopathological pathways and can assist in paving the way to future studies in the field of glaucoma, which remains to be an important silent sight-threatening disease.

The results show a weak to moderate correlation between the stability of mutated myocilin and age at glaucoma diagnosis of those patients who have the mutations. The authors should provide further explanation on the various effects of aging on protein degeneration with appropriate current references, to better explain the negative affects of aging, especially considering that the incidence of glaucoma diagnosis increases with age.   

The authors mention that one of the causes of overexpressed myocilin in TM cells is dexamethasone treatment, which can cause IOP increases. This could be an interesting mechanism to partially describe cortisone-induced glaucoma. The authors should expand on this topic and provide insights of this issue reported in literature considering the potential clinical importance of these findings, which can be also useful in a routine clinical setting.  

It is well known that diet, exercise routine, and patient lifestyle may affect IOP and influence POAG development, and can play a positive role in this disease. It would be interesting to comment further on this aspect with specific examples and current citations, with possible protective mechanisms regarding myocilin proteins.

The general clinical usefulness of genetic markers has been mentioned in the Discussion section. The authors should provide greater details on how genetic testing could be applied in a routine clinical setting and the practical measures that can be adopted in managing patients with MYOC mutations.

Author Response

We have addressed all the comments from the reviewer in the revision (highlighted in yellow).